# Effect of Antibiotics and Thermophilic Pre-Treatment on Anaerobic Co-Digestion of Pig Manure and Corn Straw

**Qinxue Wen [1], Solomon Tsegay Asfaw [1,2,*], Shuo Yang [1], Zhiqiang Chen [1], Salman Abdu Ahmed [3], Yixin Li[1] and Henok Shiferaw [4]**

[1] State Key Laboratory of Urban Water Resource and Environment (SKLUWRE), School of Environment, Harbin Institute of Technology (HIT), Harbin 150090, China; wqxshelly@163.com (Q.W.); yangshuo23@sdjzu.edu.cn (S.Y.); czqhit@163.com (Z.C.); yixinli590@gmail.com (Y.L.)

[2] School of Civil Engineering, Ethiopian Institute of Technology—Mekelle, Mekelle University, Mekelle P.O. Box 231, Ethiopia

[3] College of Power and Energy Engineering, Harbin Engineering University, Harbin 150001, China; salman2014@hrbeu.edu.cn

[4] Institute of Climate and Society, Mekelle University, Mekelle P.O. Box 231, Ethiopia; henok.shiferaw@mu.edu.et

* Correspondence: solomontsegay2023@gmail.com or solomon.tsegay@mu.edu.et

**Abstract:** The excessive use of antibiotics in the pig breeding industry leads to the accumulation of antibiotic residuals in the environment, which is attributed to the improper treatment of livestock excrements. Sulfamethoxazole (SMX) and Norfloxacin (NOR) are antibiotics used in pig breeding and veterinary medicines as growth promoters and antibacterial drugs. The aim of this study was to examine the effects of these antibiotics and thermophilic pre-treatment on methane ($CH_4$) yield by anaerobic co-digestion (AcoD) of pig manure and corn straw under mesophilic temperature condition ($35 \pm 1\ °C$). We used antibiotics at a concentration of 0, 20 and 60 mg $L^{-1}$ (three stages) in two lab-scale continuously stirred tank reactors. The first reactor was run using untreated fed and the second reactor was run with thermophilic ($55 \pm 1\ °C$) pre-treated fed. The results showed that the $CH_4$ productions from reactor one were 205, 163 and 128 and from reactor two were 222, 185 and 173 mL $CH_4$ $g^{-1}$ $VS_{added}$, respectively. This elucidates that the presence of antibiotics had a negative effect on $CH_4$ production. Moreover, thermophilic pre-treatment improved the performance of the anaerobic digestion and $CH_4$ production.

**Keywords:** anaerobic co-digestion; antibiotic; corn straw; pig manure; thermophilic pre-treatment





## 1. Introduction

China is among the world's leading countries in the rapid development of animal breeding industries [1]; hence, animal manure production has been growing. For instance, in 2018, the total livestock manure generated from large-scale centralized farms was 3190 million tons (Mt) [2], of which 793.50 Mt (28.6%) was pig manure [3]. The production of crop residue was over 800 Mt per year, of which 505.5 Mt (38.6%) was corn straw [4,5]. Meanwhile, with the development of the pig breeding industry, a wide range of veterinary antibiotics is used in pig farming for fattening, disease control and growth promotion. A high concentration of antibiotics is excreted via urine and feces as non-metabolized parent compounds [6]. Sulfamethoxazole (SMX), a member of the sulfonamides antibiotics family, and norfloxacin (NOR), a member of the quinolones antibiotics family, are the most commonly used veterinary antibiotics in the pig breeding industry [7], and they are known to be difficult to degrade in the digestive system of animals [8], as well as in the environment. Due to improper disposal of manure, antibiotic residuals with concentrations varying from 0.001 mg $kg^{-1}$ to 100 s mg $kg^{-1}$ are disposed of in the environment [9]. It has been reported by [10,11] that the maximum concentration of Norfloxacin found in pig manure was around

225.45 mg kg$^{-1}$ (WW), while Sulfamethoxazole was around 93.422 mg kg$^{-1}$ (WW) [12]. Therefore, efficient disposal of manure would be a key factor in controlling the spread of antibiotics into the environment.

Anaerobic digestion (AD) is increasingly being used as a manure management technology in animal breeding industries to treat manure and to recover energy as CH$_4$ [13]. In comparison to composting or long-term storage of manure in lagoons, AD is the most efficient pig manure management practice and a more sensitive process that requires operational precision [14]. However, the C/N ratio of pig manure is between 9.8 and 14.5, which often leads AD to inhibition of ammonia and causes an imbalance of the proton [15]. Corn straw is one of the largest agricultural residues with a high C/N ratio (from 58 to 70) [16] that contains rich cellulose and hemicellulose and has a high content of carbohydrates, about 60% of the dry matter [17]. This characteristic makes corn straw suitable for anaerobic co-digestion (AcoD) with pig manure. Furthermore, corn straw facilitates the acquisition of a high level of organic matter, as well as providing nutrients that are missing from microorganisms, which could result in a high biogas yield [18]. The presence of antibiotics in manure represents a significant concern with respect to the treatment of antibiotics during conventional agricultural waste management practices. If the concentrations of antibiotics are above a critical level, the AcoD process and its manure treatment and biogas/energy yield could be inhibited [19]. There are a lot of studies in the literature on the effects of antibiotics on CH$_4$ production [20,21]. However, few studies focus on the effects of antibiotics (SMX and NOR) on CH$_4$ production in AcoD. With the development of the livestock sector, pig farming has become one of the most important agricultural industries in China. Pig farming and pork consumption in China represent about half of the world's total [15]. To meet the growing demand of pork, Chinese farms have widely been raising swine in large numbers [22]. In this study, the AcoD of pig manure and corn straw was used to enhance the biogas production, promoting synergistic effects of microorganisms, stabilizing the digestate and increasing its amount of key nutrients [23].

Thermophilic pretreatment is a temperature-phased AcoD that promotes the hydrolysis of feedstocks and acidogenesis in the thermophilic range and ensures higher syntrophic acetogenesis and methanogenesis in the subsequent mesophilic stage [24], and is used to disintegrate the cytomembrane of the substrate on the hydrolysis process of organic compounds [25]. Ariunbaatar et al. [26] studied the possibility of enhancing the AcoD of food waste through a series of batch experiments with thermophilic pretreatment; the methane yield was increased by 40%. In another study, Zhang et al. [27] investigated the effect of thermal pretreatment on the degradation of organic compounds in food waste. They observed that thermal-pretreatment had no significant effect on the final content of protein, but decreased fat, oil and grease potential by 7–36% and increased the stagnation period of protein (35–65%). The cumulative biogas production increased linearly, and the removal efficiency of vs. and other organic matter also increased exponentially.

The main objective of this study was to assess the effect of SMX and NOR on methane yield in anaerobic co-digestion. Moreover, the thermophilic pretreated feed and the untreated feed in the same mesophilic anaerobic co-digestion were compared. The microbial dynamics of biochemical reactions such as biogas generation, VFA fractions and buffering capacity in the presence of SMX and NOR were also investigated. In this regard, the novelty of the study is the addition of antibiotics (SMX and NOR) and methane yield as an alternative energy nexus. Moreover, it supports and controls the growth and health of animals. Such a study contributes to advancing the subject field by minimizing the fears of using antibiotics in animal feeding, breeding and disease control systems.

## 2. Materials and Methods

### 2.1. Raw Materials and Inoculum

Pig manure and corn straw were collected from a local pig farm in Hulan District, Harbin, China. After sampling, the coarse particles in pig manure were screened out using a 2 mm mesh. The corn straw was naturally dried, followed by pulverization treatment.

The corn straw was cut into 2–3 cm pieces, and then ground and sieved by a 3 mm sieve mesh and subsequently pretreated by NaOH for improving biomass biodegradability and decomposition of the corn particles. Anaerobic granular sludge was obtained from a local corn processing up-flow anaerobic sludge blanket that was used as the inoculum. To enrich the methane yield and digestion microorganisms, the anaerobic sludge was activated with the corn straw powder and pig manure.

### 2.2. The Physicochemical Characteristics of Organic Materials

The total solids (TS), volatile solids (VS) and ash contents were estimated according to the standard methods of the American Public Health Association [28]. Organic loading rate (OLR) is also considered as an important operational parameter. The high OLR means high treatment capacity and methane yield, but it also may lead to overloading and thereby cause process instability; the optimal OLR used in this study was 1.68 g vs. $L^{-1}$ $d^{-1}$. The content of organic carbon was analyzed. The carbon–nitrogen ratio (25:1) was calculated using the carbon and nitrogen contents. The total solid and volatile solids were measured according to the following equations (Equations (1) and (2)). A drying oven (105 °C) and muffle furnace (600 °C) were used. The physicochemical characteristics of the corn straw and pig manure are shown in Table 1.

$$\text{Total Solids}(\%) = \left[\frac{C1 - C2}{C3 - C2}\right] \times 100\% \tag{1}$$

$$\text{Volatile Solids}(\%) = \left[\frac{(C1 - C4)}{(C3 - C2)}\right] \times 100\% \tag{2}$$

where C1 is the total wt of the dried samples and container; C2 is the wt of the container; C3 is the wt of the wet samples and container; C4 is the wt of the samples and container after ignition.

**Table 1.** Physicochemical properties of the pig manure and straw (WW).

| Substrate | TS (%) | VS (%) | MC (%) | C (%) | N (%) |
|---|---|---|---|---|---|
| Pig manure | 23.5 | 17.4 | 76.5 | 46.26 | 4.25 |
| Corn straw | 82.4 | 64.8 | 17.6 | 45.17 | 0.63 |

### 2.3. Experiment Set-Up

Two 5 L laboratory-scale reactors (CSTRs) made from Plexiglas cylinders were used. The feeding and discharge ports were set up at the top and bottom of the reactors, respectively (Figures S2 and S3 see in the Supplementary Materials). The operating temperature was maintained at 35 ± 1 °C using a temperature controller. The hydraulic retention time (HRT) of both CSTRs was maintained at around 20–25 days. Once a day, approximately 150 mL of digestate was discharged and the same amount of raw material was fed.

The co-digestion was divided into three stages: Stage I (day 0 to day 40), the reactor was fed with only a mixture of pig manure and corn straw (these 40 days included the acclimation period of the reactor); Stage II (day 41 to day 70), 20 mg $L^{-1}$ of SMX and NOR was added into the feed; Stage III (day 71–day 90), 60 mg $L^{-1}$ of SMX and NOR was added into the feed. The feed for the first reactor (R-1) was without further pretreatment. However, it passed through thermal pretreatment before being fed into the second reactor (R-2) (Figure S1 see in the Supplementary Materials). Thermal pretreatment was carried out in a 700 mL CSTR pretreatment tank with an HRT of 1–2 days and temperature of 55 ± 1 °C. The addition of the antibiotics took place periodically every day with the reactor feeding as per the feed requirements. The mixing ratio between SMX and NOR was 1:1 (Volume based).

### 2.4. Sampling and Physicochemical Analysis

Samples were collected from R-1 and R-2 approximately every two days. The sludge was immediately analyzed to measure its physicochemical characteristics, including total solids, volatile solids, pH, oxidation-reduction potential (ORP) and the concentration of soluble chemical oxygen demand (sCOD) and ammonia nitrogen ($NH_4^+$-N). The sCOD, $NH_4^+$-N, total solids and volatile solids parameters were monitored according to American Public Health Association standards. The quantification of volatile fatty acids (VFAs) (including acetic acid, propionic acid, iso-butyric acid, butyric acid, iso-valeric acid and valeric acid) was carried out using a gas chromatograph (GC7890N, Agilent, Santa Clara, CA, USA). $CH_4$ was measured by a gas chromatograph (GC7890N, Agilent, Santa Clara, CA, USA) equipped with a thermal conductivity detector and a 2 m packed column (Porapak N, Agilent, Santa Clara, CA, USA). Sulfamethoxazole and Norfloxacin were purchased from Sigma (Munich, Germany). The concentrations of the antibiotics in the anaerobic sludge and pig manure were detected and analyzed by high-performance liquid chromatography-tandem mass spectrometry (HPLC-MS/MS). The detailed information of the analysis conditions of Ultra-Performance Liquid Chromatography (UPLC) and mass spectrum was described by Wen et al. [29] All the experiments were performed in triplicate, and the results presented in this study are averages.

## 3. Results and Discussions

### 3.1. Effect of Antibiotics on Co-Digestion of Pig Manure and Corn Straw in Mesophilic AD

3.1.1. Effect on $NH_4^+$-N and sCOD on R-1

The profiles of $NH_4^+$-N in the operating stages are shown in Figure 1a. It was found that the $NH_4^+$-N concentration of Stage I averaged 751 mg $L^{-1}$ and showed higher fluctuation. In Stage II, after the introduction of the low concentration of antibiotics, the $NH_4^+$-N concentration increased from 572 to 970 mg $L^{-1}$ (with an average concentration of 763 mg $L^{-1}$). Moreover, when a low concentration of antibiotics was added in Stage II, an inhibition in the degradation of nitrogenous matter found in the pig manure was caused by the antibiotics. In Stage III, the average $NH_4^+$-N concentration increased to 883 mg $L^{-1}$. The results showed that the concentration increased by $5.71 \pm 2.38\%$ and $14.99 \pm 3.45\%$ in Stage II and Stage III, respectively, as compared to that of Stage I.

$NH_4^+$-N was mainly produced due to the degradation of proteins, nitrogenous fats and nucleic acid. According to some studies carried out on organic municipal wastes, the process remained stable and the methane yield was kept normal for $NH_4^+$-N concentrations of less than 3300 mg $L^{-1}$ [30] Moreover, Astals et al. [31] suggested that the free ammonia, which was determined by the three parameters (total ammonia concentration, temperature and pH), was the most important factor causing ammonia inhibition. $NH_4^+$-N acts as a pH neutralizer against VFAs and maintains pH at the optimum level, while, on the other hand, it is also a valuable source of nitrogen for methanogenic bacteria. On the contrary, a high concentration of ammonia can intoxicate the microorganisms and inhibit AD. According to Xiao et al. [32], pig manure usually contains excessive amounts of nitrogen, which can lead to the accumulation of $NH_4^+$-N. Moreover, the antibiotics can have negative effects on the activities of microbes, especially methanogens, when they release the $NH_4^+$-N. Overall, they co-digest into simple soluble compounds in Stage I. The maximum concentration of sCOD, i.e., 990 mg $L^{-1}$ was attained on day 6, during which the sCOD decreased with slight fluctuation in this stage. When SMX and NOR (20 mg $L^{-1}$) was introduced into the reactor in Stage II, the average daily concentration of sCOD in the effluent was 1047 mg $L^{-1}$, which was 20% higher than that in Stage I. In Stage III, the average daily sCOD concentration was 1209 mg $L^{-1}$, which was 21.56% higher than that in Stage I. The maximum sCOD accumulation (1380 mg $L^{-1}$) was observed on day 74. The sCOD concentration could be considered as a more reliable process parameter to understand the co-digestion process. Therefore, changes in the organic strength of the digesters were monitored by sCOD analysis. The inhibitory influence of antibiotics on COD removal efficiencies from the AD process has resulted in the accumulation of sCOD in the reactors. This is due to the increase

in antibiotics dosage inhibiting the activity of fermentative microorganisms or acidifying bacteria in anaerobic reactors that produce more sCOD. Moreover, the use of sCOD could be decreased due to the inhibition of antibiotics in the related activities of hydrogen-producing acetogenesis bacteria, methane-producing methanogenic agents and a combination of all these processes. The results of the research showed that the addition of SMX and NOR had a negative effect on the removal of sCOD, and this effect was recoverable. The recoverability may be due to microbial adaptation to a high concentration of the antibiotics SMX and NOR after long-term exposure. Recovery of sCOD removal was also observed by Yi et al. [33] when microbial communities were adapted to the toxic environment.

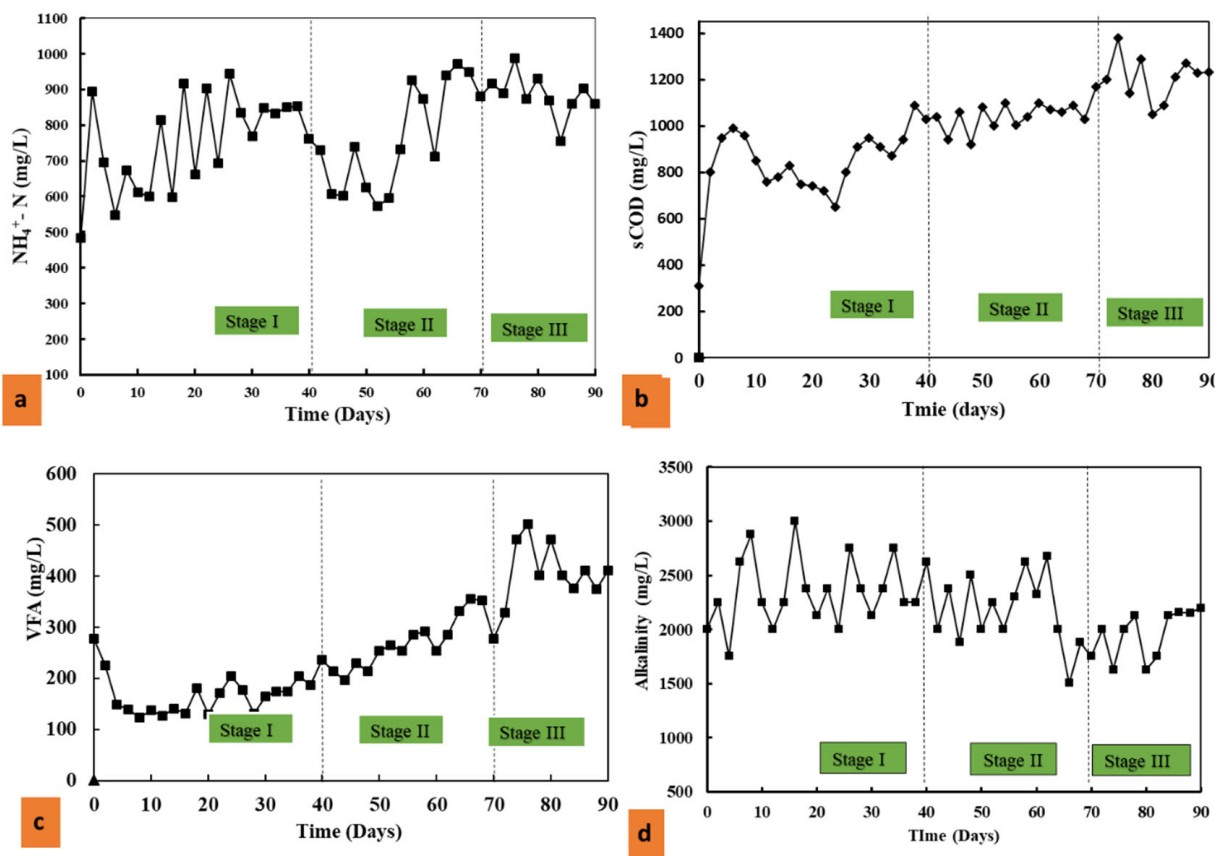

**Figure 1.** Experimental results from R-1 (**a**) $NH_4^+$-N; (**b**) sCOD; (**c**) VFAs; (**d**) total alkalinity.

3.1.2. Effects on VFAs and Alkalinity

The results of VFAs and alkalinity variation are illustrated in Figure 1c,d. VFAs and alkalinity together are recognized as good indicators of the digestion process. According to the values obtained from the gas chromatography, acetic acid occupied the majority of the VFAs' composition in the reactor. Small amounts of propionic and butyric acid were also detected.

During the beginning of the AD process (day 1), the concentration of VFAs rose up to 278 mg $L^{-1}$ due to the overloading of organic materials. On the subsequent day (day 2), it began to decrease and fluctuated between 120 to 240 mg $L^{-1}$, with an average concentration of 176 mg $L^{-1}$ (Figure 1c). The alkalinity in the reactor ranged between 1400 and 3000 mg $L^{-1}$ and averaged 2353 mg $L^{-1}$. The ratio of VFAs/alkalinity in the reactor was 0.07 during this stage. The VFAs in Stage II were on average 271 mg $L^{-1}$, which was 35.03% higher than that in Stage I. The alkalinity was 2138 mg $L^{-1}$, and the VFAs/alkalinity ratio was 0.20. The average concentration of VFAs was 415 mg $L^{-1}$ in Stage III, which was higher by 57.67% and 34.75% than those in Stage I and Stage II, respectively (Table S2 see in the Supplementary Materials). As the concentration of the antibiotics in the

reactor increased, the resistance of microorganisms to the antibiotics also increased due to prolonged exposure to antibiotics. A significant accumulation of VFAs was observed in this stage. The total alkalinity ranged from 1000 to 2000 mg $L^{-1}$ with an average value of 1936 mg $L^{-1}$, which was slightly lower than that in Stage II, but still satisfied the requirement of AD. The VFAs/Alkalinity ratio in this stage is 0.21. The results indicated that the ratio of VFAs to alkalinity did not affect the performance of anaerobic co-digestion. According to the study carried out by Razaviarani et al. [34], a ratio of total acidity/alkalinity between 0.1 and 0.4 indicates favorable operating conditions without the risk of acidification. This indicates that the presence of SMX and NOR in AD had little effect on VFAs and alkalinity profiles.

### 3.1.3. The Effects of Antibiotics on Methane Yield

The $CH_4$ production profiles in all the operation stages are shown in Figure 2. The average daily $CH_4$ production in Stages I, II and III were found to be 196, 149 and 128 mL $CH_4$ $d^{-1}$, respectively. The $CH_4$ production fluctuated between 160 and 238 mL $CH_4$ $g^{-1}$ $VS_{added}$ in Stage I and between 100 and 210 mL $CH_4$ $g^{-1}$ $VS_{added}$ in Stage II, respectively. At day 54, the trend of $CH_4$ production significantly declined (to around 101 $CH_4$ mL $g^{-1}$ $VS_{added}$) and started to recover due to the adaptation of microorganisms to the low concentration of antibiotics; the average $CH_4$ production in this stage was 163 mL $CH_4$ $g^{-1}$ $VS_{added}$, which was 23.83% lower than that in Stage I.

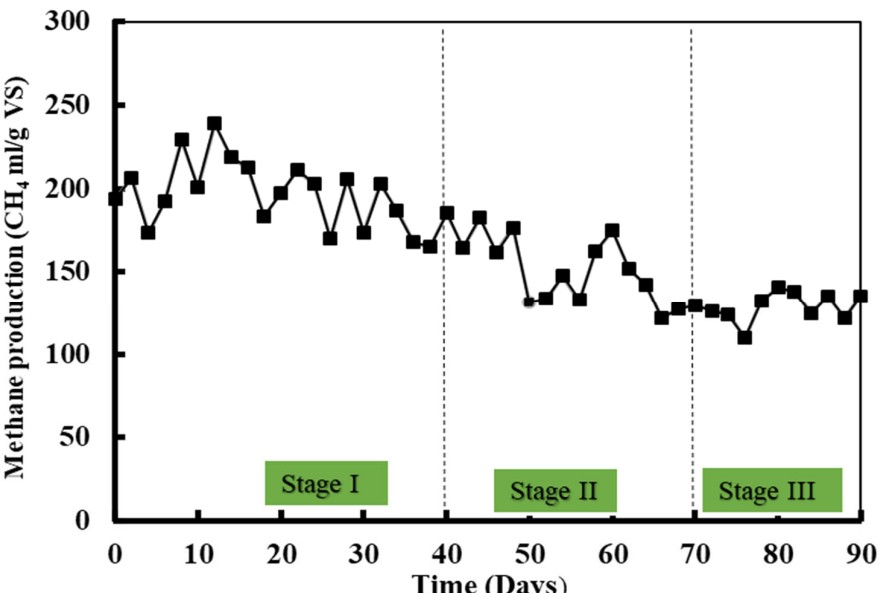

**Figure 2.** R-1 $CH_4$ production change.

In Stage III, $CH_4$ production was in the range of 105–140 mL $CH_4$ $g^{-1}$ $VS_{added}$. The $CH_4$ production in the reactor declined to about 110 mL $CH_4$ $g^{-1}$ $VS_{added}$ (day 76) and maintained with a slight fluctuation. The average $CH_4$ production in this stage was 128 mL $CH_4$ $g^{-1}$ $VS_{added}$, which was 34.31% lower than that in Stage I. The results showed that the presence of antibiotics had a negative effect ($p < 0.05$) on biogas and $CH_4$ production. Moreover, the inhibitory effect of antibiotics' $CH_4$ production increases with the increase in antibiotics' concentration.

### 3.1.4. Antibiotics Removal Efficiency on R-1

The levels of SMX and NOR in the digestate in R-1 in Stages II and III were measured to reveal their behavior in an anaerobic digester. The concentrations of SMX and NOR were measured on day 64 and day 88 to determine the removal efficiency of the antibiotics under the mesophilic condition. The antibiotic removal efficiency during the anaerobic co-digestion is shown in Table 2. The concentrations of SMX and NOR in R-1 on day 88

were three times higher than those on day 64. This was due to the continuous addition of high concentrations of the antibiotics, although the removal efficiencies of SMX and NOR on day 88 were 2.46% and 4.60% higher than those on day 64, respectively. Xiao. et al. [32] and Qiu et al. [35] showed that antibiotics can be removed by sorption and biodegradation due to sulfur-reducing bacteria.

**Table 2.** The removal efficiency of SMX and NOR.

| Reactor | Sampling Day | Antibiotics Type | Antibiotics Concentration (mg L$^{-1}$) | Removal Efficiency (%) |
|---|---|---|---|---|
| R-1 | 64 | SMX | $5.81 \pm 0.115$ | $71.10 \pm 0.58$ |
| | | NOR | $7.846 \pm 0.234$ | $60.79 \pm 1.17$ |
| | 88 | SMX | $15.864 \pm 0.215$ | $73.56 \pm 1.08$ |
| | | NOR | $20.768 \pm 0.156$ | $65.39 \pm 0.75$ |
| R-2 | 64 | SMX | $4.523 \pm 0.155$ | $72.19 \pm 0.69$ |
| | | NOR | $5.79 \pm 1.22$ | $73.28 \pm 0.59$ |
| | 88 | SMX | $11.428 \pm 0.201$ | $77.39 \pm 0.78$ |
| | | NOR | $16.681 \pm 0.137$ | $80.95 \pm 1.01$ |

### 3.2. The Effects of Thermophilic Pre-Treatment on the Co-Digestion of Pig Manure and Corn Straw

#### 3.2.1. Effects on NH4+-N and sCOD

Figure 3a shows the daily $NH_4^+$-N concentration of R-2. As seen in the figure, $NH_4^+$-N was on average 737, 662 and 698 mg L$^{-1}$ in each stage, respectively. By comparison, R-1 decreased by 1.8%, 15.255% and 29.59%, respectively. The decrease in $NH_4^+$-N is attributed to the thermophilic pretreatment leading to improved decomposition of nitrogen-containing substances [36]. In addition, thermophilic pretreatment may disintegrate the solid particles of pig manure; as a result, more soluble substances were subsequently released, which can be more easily utilized by microorganisms, reducing the concentration of $NH_4^+$-N. The concentration was lower in R-2 than in R-1; hence, enhanced protein fermentation was observed under thermophilic conditions [37]. The variation of sCOD is shown in (Figure 3b). The average sCOD concentrations in R-2 were 472, 881 and 1100 mg L$^{-1}$ in each stage, respectively, which were 45.15%, 18.80% and 15.20% lower compared to those in R-1. The results indicated that the continuous hydrolysis of particulates resulted in more soluble products being released during the co-digestion. Such conversions increased the amount of soluble organic matter that was accessible to the methanogenic bacteria for $CH_4$ production [38].

#### 3.2.2. Effects on VFAs and alkalinity

The average VFAs content in R-2 was 175, 252 and 345 mg L$^{-1}$, respectively, in the three operation stages (Figure 3c). These were 8.76, 7.14 and 20.34% lower than those in R-1, respectively. The accumulation reduction of VFAs in R-2 occurred due to the thermal pretreatment influence in the influent characteristics. This indicates that acetogens and hydrogenotrophs under thermophilic pretreatment conditions are more sensitive to environmental changes [39].

The average alkalinity in Stages I, II and III in R-2 was found to be 2552, 1834 and 1756 mg L$^{-1}$, respectively (Figure 3d). Compared to those of R-1, it decreased by 7.8, 16.56 and 10.25% respectively. The result indicated that the thermophilic pretreatment increased the buffering capacity of the system. The VFAs/Alkalinity ratio in R-2 was 0.06, 0.19 and 0.21 in Stages I, II and III respectively. Cetecioglu et al. [39] reported that the buffering capacity was sufficient when the VFAs/alkalinity ratio was maintained below 0.4.

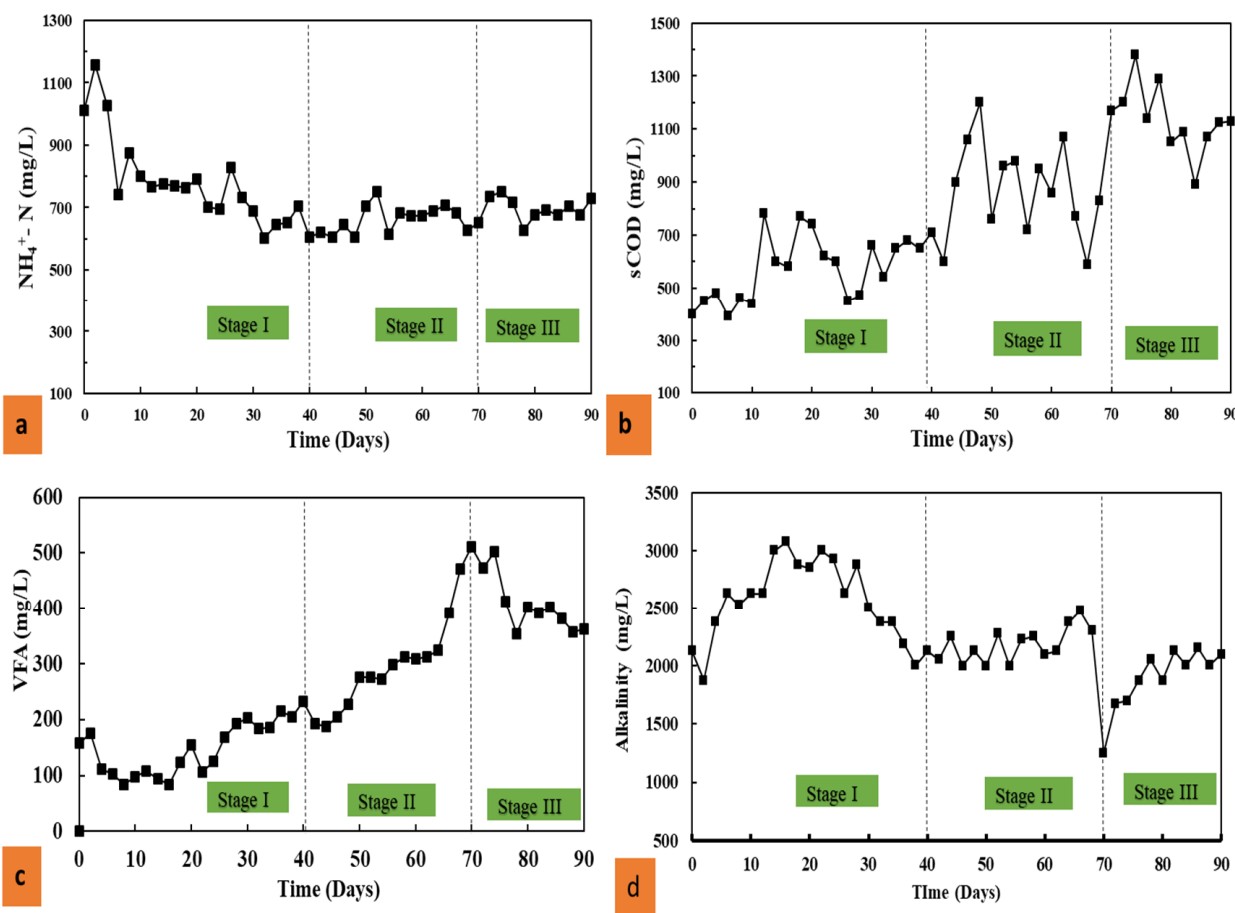

**Figure 3.** Experimental results from R-2 (**a**) $NH_4^+$-N; (**b**) sCOD; (**c**) VFAs; (**d**) total alkalinity.

### 3.2.3. Effects on Methane Production

The average $CH_4$ production in Stage I was 222 mL $CH_4$ $g^{-1}$ $VS_{added}$ (Figure 4) in R-2. By comparison, in R-1, $CH_4$ production increased by 8.3%, which was due to thermophilic pretreatment promoting the hydrolysis of the substrate, increasing the $CH_4$ production. The average $CH_4$ production in Stage II was 185 mL $CH_4$ $g^{-1}$ $VS_{added}$, which was 13.3% higher than that of R-1. In Stage III, the average $CH_4$ production was 173 mL $CH_4$ $g^{-1}$ $VS_{added}$, which was 35.3% higher than that of R-1. This was because the thermophilic temperature accelerated the degradation of antibiotics and facilitated the reproduction of the microbial metabolisms, thus enhancing the $CH_4$ production efficiency of the system. Stages II and III had similar $CH_4$ production even though the concentration of antibiotics in Stage III was triple that of Stage II (185 and 173 mL $CH_4$ $g^{-1}$ $VS_{added}$ respectively) (Figure 5).

Figure 5 showed the comparison of $CH_4$ production between R-1 and R-2. The $CH_4$ production from R-2 was increased by 11.26%, 9.44% and 26.01% in each stage, respectively, from that of R-1. This is due to thermophilic pretreatment increasing the buffering capacity of the system and improving the $CH_4$ production. Moreover, thermo-pretreatment can reduce the inhibitory effect of antibiotics and also improve the degradation of antibiotics in the AcoD system. Our results have similarities with those of other studies. For example, studies by Li et al. [40] and Zhi et al. [41] found a reduced inhibitory effect and improved degradation of antibiotics when they used thermo-pretreatment. Moreover, since pig manure and corn straw contain lignocellulosic biomasses that are resistant to hydrolysis [42], due to the presence of recalcitrant substances such as lignin thermophilic, pretreatment would enhance the hydrolysis of lignocellulosic biomass and improve the digestion of organic substrates.

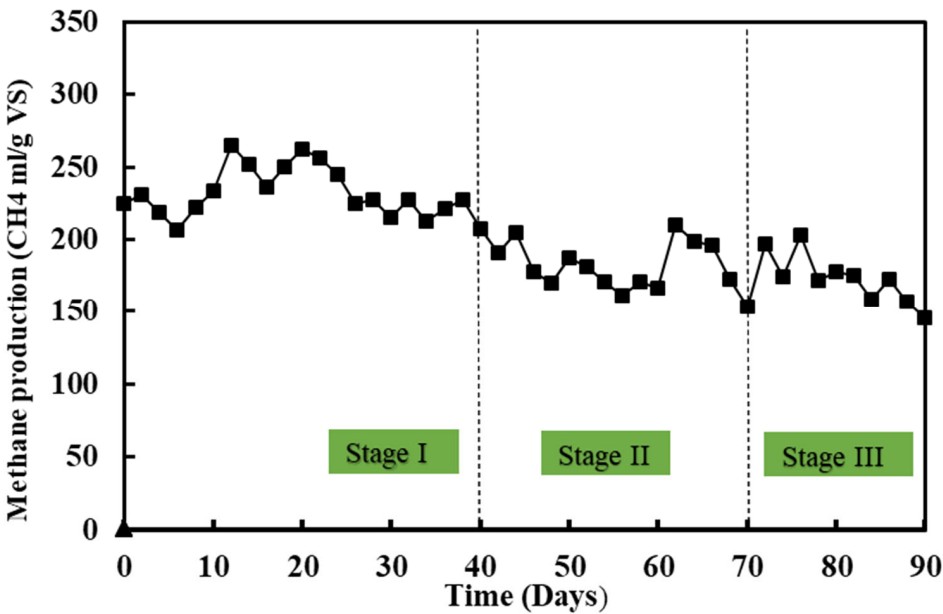

**Figure 4.** R-2 CH$_4$ production change.

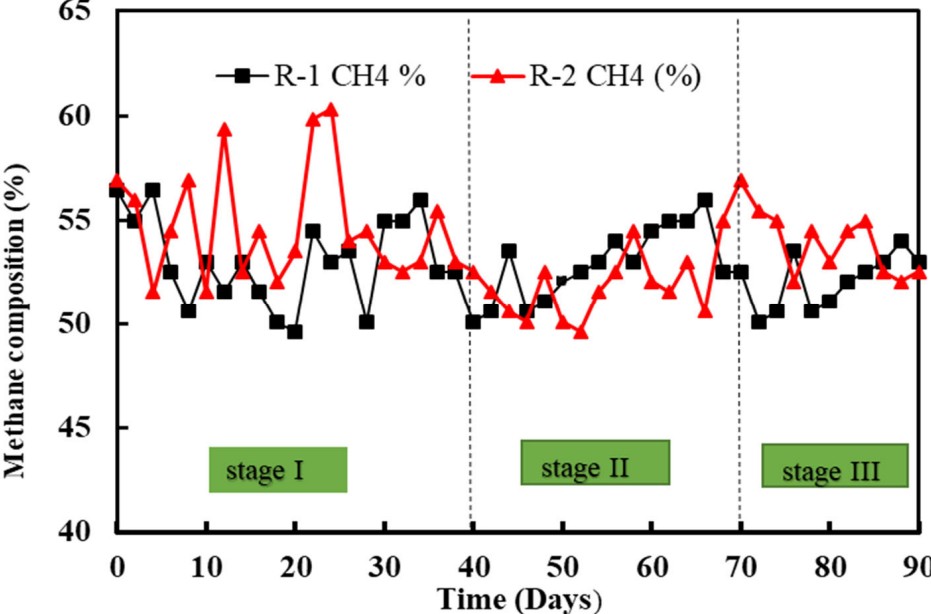

**Figure 5.** CH$_4$ percentage composition in R-1 and R-2.

### 3.2.4. Antibiotics Removal Efficiency on R-2

The removal of antibiotics in R-2 is shown in Table 2. The antibiotics' concentrations were measured on day 64 and day 88, respectively. The results showed that, compared to R-1, the removal efficiency of SMX increased by 1.6 ± 0.28% (day 64) and 4.74 ± 0.95% (day 88), and the removal efficiency of NOR increased by 11.69 ± 0.76% and 10.95 ± 0.93%, respectively. Despite the difference in concentration, thermophilic pretreatment enhanced the removal efficiency of both types of antibiotics. Therefore, more stable antibiotics removal could have been achieved for higher antibiotics concentrations when thermophilic pretreatment was used.

## 4. Conclusions

The performance of the AD process treatment of pig manure and corn straw under different SMX and NOR concentrations was investigated. Moreover, the effect of thermophilic pretreatment was investigated. and we found the following:

1.  The addition of SMX and NOR had a negative effect on the digestion process;
2.  The CH$_4$ production decreased by 20.48% in stage two and 37.56% in stage three;
3.  Thermophilic pretreatment increased the buffering capacity of the system and improved methane production by 15.57%;
4.  Thermal pretreatment also improved the degradation of SMX by 1.60% and 4.75% and NOR by 11.67% and 10.95% in the low and high antibiotic concentration stages;
5.  The presence of the antibiotics affects the CH4 production.

**Supplementary Materials:** The following supporting information can be downloaded at: https://www.mdpi.com/article/10.3390/w15183223/s1, Figure S1: Schematic description of mixing and anaerobic Co-digestion on CSTR used for anaerobic co-digestion of pig manure and corn straw in reactor-1 with untreated; Figure S2: Schematic description of CSTR used for anaerobic co-digestion of pig manure and corn straw in reactor-1 with untreated feeding; Figure S3: Schematic description of CSTR used for anaerobic co-digestion of pig manure and corn straw in reactor-2 with thermophilic pretreatment feeding; Figure S4: Experimental results between each stage and R-1and R-2.

**Author Contributions:** Conceptualization, experimental work, and drafting the manuscript, S.T.A.; Provided guidance and reviews, Q.W. and Z.C.; Assisted experiments Y.L.; Project administration, S.Y.; Validation and graphics, S.A.A.; Review and editing, H.S. All authors have read and agreed to the published version of the manuscript.

**Funding:** The authors acknowledge the financial support of the National Key R&D Program of China (2018YFC0406303, 2016YFC04011023), the National Natural Science Foundation of China (Grant No. 51878214) and National and local joint engineering laboratory of municipal sewage resource utilization technology (2018KF06) and UTFORSK 2016—Long-term project funding from Norway (UTF-2016-longterm/10042).

**Data Availability Statement:** The data and material used in this research are available upon request from the corresponding author.

**Conflicts of Interest:** No potential conflict of interest was reported by the authors.

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
