# Peer review of "Effect of Antibiotics and Thermophilic Pre-Treatment on Anaerobic Co-Digestion of Pig Manure and Corn Straw"

_water, doi:10.3390/w15183223_

Round 1
Reviewer 1 Report
The article investigated the effects of antibiotics and sulfamethoxazole and norfloxacin on anaerobic co-digestion of pig manure and corn straw. Although there are several studies on the performance evaluation of anaerobic digestion with antibiotic addition, less is known about the addition of SMX and NOR antibiotics along with thermophilic pretreatment. Therefore, the findings are considered interesting. The paper is well-written and east to follow. This is a study that could potentially add significantly to current knowledge on the effects of antibiotic addition on methane production from anaerobic digestion. The methods include all necessary elements and are very well described. The discussion is articulated well but could be improved. A few points to include are the novelty of the study and how this study could potentially advance the field. Overall, the topic of this article is extremely important since antibiotic resistance accumulation in the environment is an emerging public health concern.
Author Response
Dear Reviewer,
Thank you for your valuable comments and suggestions you contributed for enhancement of the manuscript. We really have appreciated for the technical and editorial comments. The comments and suggestions are really constructive and significantly helped us to improve the quality and standard of the manuscript. We tried to go through the comments given point by point here under below. We believe now carefully addressed the comments and suggestions and come up with the updated manuscript. The changes are highlighted in yellow color in the main document. We hope that you will be satisfied by our revision of the updated of the manuscript.

Reviewer 2 Report
The effect of antibiotic metabolites in the manure on methane production and VFAs can be included.
Minor edits in line number 16, 94 and 250 needed
Quality of English used is generally good but use of compound sentences in the results and discussion can be avoided.
Author Response

(The authors gave the same response as above.)

Reviewer 3 Report
The authors present the results of research on the effect of the antibiotics Sulfamethoxazole (SMX) and Norfloxacin (NOR), which are used in pig farming and in veterinary medicine as growth stimulants and antibacterial drugs, and thermophilic pretreatment on the yield of methane (CH4) during anaerobic co-digestion (AcoD) of pig manure and corn straw under mesophilic conditions. The research was carried out in three stages in two continuous stirred laboratory reactors. One reactor was operated as reference without thermophilic pre-treatment. In this reactor, the effect of antibiotics on the process of co-digestion of pig manure with corn straw in mesophilic conditions was investigated. The research results show that the addition of antibiotics has a negative effect on sCOD. However, this effect can be eliminated by adapting to higher concentrations of antibiotics. The presence of antibiotics showed little effect on the production of VFAs and the alkalinity of the process. However, the presence of antibiotics negatively affected biogas and CH4 production, while the inhibitory effect increased with increasing antibiotic concentration. The results of the thermophilic pretreatment investigation (2nd DG bioreactor) showed an increase in the buffering capacity of the system and an increase in methane production. Heat pretreatment at low and high concentration of antibiotics also improved their degradation.
Author Response

(The authors gave the same response as above.)
